# Interplay between OmpA and RpoN Regulates Flagellar Synthesis in *Stenotrophomonas maltophilia*

**DOI:** 10.3390/microorganisms9061216

**Published:** 2021-06-04

**Authors:** Chun-Hsing Liao, Chia-Lun Chang, Hsin-Hui Huang, Yi-Tsung Lin, Li-Hua Li, Tsuey-Ching Yang

**Affiliations:** 1Division of Infectious Disease, Far Eastern Memorial Hospital, New Taipei City 220, Taiwan; liaochunhsing@gmail.com; 2Department of Medicine, National Yang Ming Chiao Tung University, Taipei 112, Taiwan; ytlin8@vghtpe.gov.tw; 3Department of Biotechnology and Laboratory Science in Medicine, National Yang Ming Chiao Tung University, Taipei 112, Taiwan; kartd0087603@gmail.com (C.-L.C.); toe3273917@outlook.com (H.-H.H.); 4Division of Infectious Diseases, Department of Medicine, Taipei Veterans General Hospital, Taipei 112, Taiwan; 5Department of Pathology and Laboratory Medicine, Taipei Veterans General Hosiptal, Taipei 112, Taiwan; lilh@vghtpe.gov.tw; 6Ph.D. Program in Medical Biotechnology, Taipei Medical University, Taipei 110, Taiwan

**Keywords:** *Stenotrophomonas maltophilia*, OmpA, swimming, flagellum

## Abstract

*OmpA*, which encodes outer membrane protein A (OmpA), is the most abundant transcript in *Stenotrophomonas maltophilia* based on transcriptome analyses. The functions of OmpA, including adhesion, biofilm formation, drug resistance, and immune response targets, have been reported in some microorganisms, but few functions are known in *S. maltophilia*. This study aimed to elucidate the relationship between OmpA and swimming motility in *S. maltophilia*. KJΔOmpA, an *ompA* mutant, displayed compromised swimming and failure of conjugation-mediated plasmid transportation. The hierarchical organization of flagella synthesis genes in *S. maltophilia* was established by referencing the *Pseudomonas aeruginosa* model and was confirmed using mutant construction, qRT-PCR, and functional assays. Distinct from the *P. aeruginosa* model, *rpoN*, rather than *fleQ* and *fliA*, was at the top of the flagellar regulatory cascade in *S. maltophilia*. To elucidate the underlying mechanism responsible for *ΔompA*-mediated swimming compromise, transcriptome analysis of KJ and KJΔOmpA was performed and revealed *rpoN* downregulation in KJΔOmpA as the key element. The involvement of *rpoN* in *ΔompA*-mediated swimming compromise was verified using *rpoN* complementation, qRT-PCR, and function assays. Collectively, OmpA, which contributes to bacterial conjugation and swimming, is a promising target for adjuvant design in *S. maltophilia*.

## 1. Introduction

Gram-negative bacteria are surrounded by two membranes, which confine the periplasmic space containing the peptidoglycan [1]. This double membrane may be impenetrable to antibiotics, noxious agents, or other foreign compounds [2,3]. The outer membrane (OM) is a complex organelle that provides a barrier to protect bacteria from hazards in their environment. The OM of Gram-negative bacteria is unique in its composition and shows asymmetrical lipid distribution. Approximately 50% of the OM mass consists of proteins, either integral membrane proteins or membrane-anchored lipoproteins. Thus, as much as 3% of the Gram-negative bacterial genome may encode outer membrane proteins (OMPs) [4]. Nearly all of the integral membrane proteins of the OM assume a β-barrel architecture with long loops between the strands on the extracellular side and short loops on the periplasmic side. β-barrel OMPs are classified into two major types, classical porins and slow porins, based on their physiological roles. Classical porins serve as ports of entry for hydrophilic molecules in a non-selective fashion, facilitating the diffusion of nutrients and extrusion of waste products [5]. OmpF and OmpC are the most studied classical porins in Gram-negative bacteria and are highly conserved throughout the Enterobacterales [6,7]. In contrast, slow porins have very low permeability, and molecular transportation is not their main function. *Escherichia coli* OmpA, *Acinetobacter baumannii* OmpA, and *Pseudomonas aeruginosa* OprF are examples of slow porins. OmpA is a surface-exposed porin protein with a high copy number among the OMPs of *E. coli* [8]. Its N-terminal domain comprises eight transmembrane antiparallel β-strands, and its C-terminal domain interacts with the peptidoglycan layer [9,10]. OmpA is involved in adhesion, invasion, biofilm formation, drug resistance, and bacteriophage entry [11,12,13,14]. In addition, OmpA acts as an immune target and induces host immune responses [15,16]; thus, it is the most popular vaccine candidate widely developed in *E. coli*, *Klebsiella pneumoniae*, and *A. baumannii* [17,18,19]. 

Swimming motility is a critical aspect of bacterial pathogenesis and is required for host colonization. It is also crucial for many biological functions including nutrient acquisition, stress avoidance, and sexual reproduction [20,21]. The flagellum is the most important organelle for bacterial swimming. It is composed of a basal body, hook, and a long, thin filament. The basal body, consisting of rotors and a stator, is embedded in the bacterial membrane. The hook is a joint that connects the basal body and filament. Flagellin is the subunit protein of the filament, which is a whip-like appendage that enables bacterial motility. Bacterial flagellum assembly requires the expression of nearly 60 genes in multiple operons, which are clustered at several loci on the chromosome [22]. Studies on flagellum synthesis have been carried out in several bacteria, including *E. coli*, *Salmonella typhimurium* [23], *Caulobacter crescentus* [24], *Vibrio cholerae* [25], *V. parahaemolyticuss* [26], and *P. aeruginosa* [27]. Flagellum synthesis has been hierarchically organized. A transcription of flagellum synthesis genes forms an ordered cascade whereby a gene located at a higher level must first be expressed before that of the next level. In the hierarchical cascade, several checkpoints are set to ensure that flagellar components are expressed and assembled sequentially. Flagellum synthesis is hierarchically regulated at three levels in *E. coli* [28] and is hierarchically classified into four classes in *C. crescentus* [24], *V. cholerae* [25], and *P. aeruginosa* [27]. In addition to a functional flagellum, effective swimming relies on the bacterial repertoire for sensing environmental conditions. Thus, a pathogen generally harbors several efficient signaling systems to sense environmental stimuli or quorum sensing (QS) signals, direct their movement to attractants, and avoid repellents.

*Stenotrophomonas maltophilia*, widely distributed in nature and in hospitals, has emerged as a global opportunistic Gram-negative pathogen, especially in patients with cystic fibrosis and immunocompromised individuals [29]. *S. maltophilia* is considered to be a low-virulence pathogen; however, the treatment of an *S. maltophilia* infection is a major challenge due to its intrinsic resistance to several antibiotics that are commonly used in clinics [30]. The reported resistance mechanisms in *S. maltophilia* include poor membrane permeability, antibiotic hydrolysis or modification, and efflux pump overexpression [31]. Thus, it is increasingly challenging for physicians to use conventional therapies to effectively treat *S. maltophilia* infections [32]. Novel convincing targets in *S. maltophilia* are greatly needed for the development of antibiotics, adjuvants, and even vaccines. Transcriptome analysis of *S. maltophilia* KJ was performed in our recent study, and the results demonstrated that *ompA* (Smlt0955) has the greatest transcript abundance in logarithmically grown KJ cells [33]. Given our understanding of OmpA in other Gram-negative bacteria, the OmpA of *S. maltophilia* is a prospective candidate to be first considered for the development of antibiotics, adjuvants, or vaccines. However, the current understanding of OmpA in *S. maltophilia* is relatively deficient, except for the reports by Li et al. that focused on the immunogenic properties of OmpA and the host protective immune response [34,35]. Given that OmpA is the most abundant OMP in *S. maltophilia*, it should be a critical contributor to the maintenance of membrane integrity. Bacterial membrane integrity has a significant impact on flagellum assembly and swimming motility. In this study, the link between OmpA and swimming motility in *S. maltophilia* was investigated. We revealed that the deletion of *ompA* resulted in swimming compromise and thus conducted experiments aimed at elucidating the underlying molecular mechanism.

## 2. Materials and Methods

### 2.1. Bacterial Strains, Media, Plasmids, and Primers 

The bacterial strains, plasmids, and primers used in this study are listed in Appendix A. *S. mlatophilia* KJ was an isolate isolated from sputum and identified using the Phoenix^TM^100 system (Becton Dickinson).

### 2.2. Construction of in-Frame Deletion Mutants

The in-frame deletion mutants were constructed using double cross-over homologous recombination. The DNA fragments flanking the deleted genes were amplified using PCR and subsequently cloned into pEX18Tc to generate the mutagenic plasmids. The primers set for the construction of mutagenic plasmids were OmpAN-F/OmpAN-R and OmpAC-F/OmpAC-R for pΔOmpA, FliC1N-F/FliC1N-R and FliC3C-F/FliC3C-R for pΔFliC1C2C3, RpoNN-F/RpoNN-R and RpoNC-F/RpoNC-R for pΔRpoN, FleQN-F/FleQN-R and FleQC-F/FleQC-R for pΔFleQ, as well as FliAN-F/FliAN-R and FliAC-F/FliAC-R for pΔFliA (Appendix A). Mutagenic plasmids were transferred into *S. maltophilia* by conjugation. The plasmid’s conjugation, transconjugant’s selection, and mutant’s confirmation were carried out as described previously [36]. 

### 2.3. Construction of KJL2::OmpAΔOmpA and KJL2::RpoNΔOmpA

As plasmid transportation by conjugation is unavailable in the *ompA* mutant (KJΔOmpA), an alternative strategy was designed for gene expression in KJΔOmpA. First, the gene intended to be expressed in KJΔOmpA was chromosomally replaced with the *L2* gene using double cross-over homologous recombination; the prototype chromosomal *ompA* gene was then deleted. For this purpose, we first constructed a recombinant plasmid, pEXCJ1. The 353-bp and 376-bp DNA fragments upstream and downstream of the *L2* gene were obtained by PCR using primers pEXCJ1N-F/pEXCJ1N-R and pEXCJ1C-F/pEXCJ1C-R (Appendix A), respectively, and were subsequently cloned into pEX18Tc to yield pEXCJ1. The multiple cloning sites (SphI/PstI/SalI/XbaI/BamHI/SmaI/KpnI/SacI) of pEX18Tc were used for cloning the exotic gene intended to be expressed in KJΔOmpA. 

The intact *ompA* and *rpoN* genes were amplified by PCR using the primer sets OmpA-F/OmpA-R and RpoN-F/RpoN-R (Appendix A), respectively, and were then cloned into pEXCJ1 to yield pCJ1-OmpA and pCJ1-RpoN. The plasmids pCJ1-OmpA and pCJ1-RpoN were transferred into *S. maltophilia* KJ to generate KJL2::OmpA and KJL2::RpoN, respectively. The prototype chromosomal *ompA* gene was deleted from KJL2::OmpA and KJL2::RpoN to yield KJL2::OmpAΔOmpA and KJL2::RpoNΔOmpA, respectively. 

### 2.4. Construction of P_rpoN_-_xylE_ Transcription Fusion Plasmid, pRpoN_xylE_

The 576-bp DNA fragment upstream of the *rpoN* gene was obtained by PCR using primer sets RpoNN-F/RpoNN-R (Appendix A) and cloned into pRKxylE, a *xylE* reporter plasmid, to generate pRpoN_xylE_. The plasmid pRpoN_xylE_ was mobilized into *S. maltophilia* strains indicated for a promoter activity assay.

### 2.5. OMP Preparation and SDS-PAGE

Logarithmic-phase cells were harvested and OMPs were prepared using the *N*-lauroylsarcosine method described previously [37]. The purified OMPs were boiled for 5 min at 100 °C and then separated using SDS-PAGE electrophoresis in a 15% polyacrylamide gel. The gel was stained with 0.1% Coomassie brilliant blue R250 (Bio-Rad) and de-stained with 40% methanol/10% glacial acetic acid for the visualization of proteins.

### 2.6. Catechol 2,3-Dioxygenase (C23O) Activity Assay

Catechol-2,3-dioxygenase is encoded by the *xylE* gene, and its activity was measured using 100 mM catechol as the substrate, as described previously [38]. The hydrolysis rate of catechol was calculated using 44,000 M^−1^cm^−1^ as the extinction coefficient. One unit of enzyme activity (U) was defined as the amount of enzyme that converts 1 nmole of substrate per minute. The specific activity was expressed as U/OD_450_ nm.

### 2.7. Swimming Assay

Five µL of a logarithmically grown bacterial cell culture were inoculated onto the swimming agar (1% tryptone, 0.5% NaCl, and 0.15% agar). The plates were incubated at 37 °C for 48 h and the diameters of the swimming zones (mm) were recorded. 

### 2.8. Flagella Staining 

We analyzed the presence of flagella using negative staining with 1% phosphotungstic acid (pH 7.4) and observed using transmission electron microscopy (TEM) (Hitachi H-7650 microscope), as described previously [39].

### 2.9. Preparation of Polyclonal Anti-Rabbit Anti-FliC3 Antibody

Of the three *fliC* homologue genes, *fliC3* (Smlt2304) is annotated as *fliC* in the *S. maltophilia* K279a genome [40]; thus, it was selected as the candidate for the preparation of a polyclonal antibody. The full coding sequence of *fliC3* was amplified by PCR using the primer sets FliC3his-F and FliC3his-R (Appendix A) and ligated to the vector pET-24b, yielding pETFliC3. For expression, the recombinant plasmid pETFliC3 was transformed into *E. coli* BL21(DE3). The logarithmically grown *E. coli* BL21(DE3)(pETFliC3) cells were treated with 1 mM IPTG for 3 h and the fusion proteins FliC3-6His were purified with Ni-NTA resin. Antibodies against the FliC3-6His protein were raised by the immunization of New Zealand rabbits with the purified FliC3-6His protein.

### 2.10. SDS–PAGE and Western Blot Analysis

The logarithmically grown bacterial cells were disrupted via sonication. Intracellular proteins were separated using electrophoresis in 15% (*w*/*v*) SDS–PAGE, transferred onto polyvinylidene difluoride (PVDF) membranes, and probed with anti-FliC3 and anti-RpoA antibodies. The protein RpoA served as an intracellular control. 

### 2.11. Transcriptome Analysis 

Overnight cultures of KJ and KJΔOmpA were subcultured into fresh LB at an initial OD_450_ nm of 0.15 and incubated for 5 h to a logarithmic phase. Total RNA isolation, rRNA depletion, adapter-ligated cDNA library construction and enrichment, and cDNA sequencing were performed as described previously [41]. The sequencing reads were mapped to the *S. maltophilia* K279a genome. RNA-seq results were analyzed using a CLC Genomics Workbench v 6.0 (CLC Bio) and presented as Reads Per Kilobase per Million mapped reads (RPKM).

### 2.12. Quantitative Reverse Transcription-PCR (qRT-PCR)

DNA-free RNA for qRT-PCR was prepared from logarithmically grown bacterial cells, as described previously [42]. The primers used for the qRT-PCR of each gene are listed in Appendix A. The relative transcription level was calculated using the comparative *C_T_* (ΔΔ*C_T_*) method [43] and 16S rRNA was used as the endogenous control. Each result represents the average of three independent determinations.

## 3. Results

### 3.1. Bioinformatics Analysis of OmpA

The transcriptome analysis of KJ cells in the logarithmic phase was performed in our recent study [33]. *OmpA* (Smlt0955) showed the greatest transcript abundance in wild-type KJ under our experimental conditions. The presence of any protein in such a large amount suggests that it may play a significant role in bacteria. Based on this, we investigated the functions of OmpA and proposed some innovative aspects for this OMP. 

*OmpA*, encoding a 366-amino acid β-barrel OMP, is well conserved in all of the sequenced *S. maltophilia* genomes. BLAST analysis showed that OmpAs of *S. maltophilia* strains shared 86 to 99% protein identities. However, *S. maltophilia* OmpA showed relatively lower identity with its homologues in other bacterial species, such as 22% identity with the OmpA of *E. coli* and 32% identity with the OprF of *P. aeruginosa* PAO1.

### 3.2. OmpA Deletion Impairs Bacterial Conjugation

To elucidate the function of OmpA in *S. maltophilia*, an in-frame deletion mutant of *ompA*, KJΔOmpA, was constructed by deleting the OmpA-like domain (amino acids 297–357) (Figure 1). A recombinant plasmid, pOmpA, containing an intact *ompA* gene was prepared for the complementary assay of KJΔOmpA. Despite several attempts to introduce pOmpA back into KJΔOmpA by conjugation, no transconjugant was obtained. Our attempts to transfer the empty plasmid, pRK415, into KJΔOmpA by conjugation also failed. In contrast, the plasmid pOmpA was successfully transferred into wild-type KJ, yielding KJ(pOmpA). Collectively, the failure of the conjugation-mediated plasmid transfer appeared to originate from the lost OmpA function, implying that OmpA is a critical OMP for conjugation in *S. maltophilia*.

An alternative strategy was applied for the complementary assay. We first generated a conditional construct of *S. maltophilia* KJ, KJL2::OmpA, in which the chromosomal *L2* gene was replaced with an intact *ompA* gene (Figure 1). Thus, in this construct, the *ompA* gene was driven by the *L*2 promoter, *P_L_*_2_. *P_L_*_2_ has basal-level activity in the absence of β-lactam and can be upregulated upon β-lactam challenge [44]. Next, the prototype *ompA* gene in KJL2::OmpA was deleted via a double cross-over homologous recombination to yield KJL2::OmpAΔOmpA (Figure 1), which was regarded as an alternative complementary strain of the mutant KJΔOmpA. The OMP profiles of wild-type KJ, mutant KJΔOmpA, and the complementary strain KJL2::OmpAΔOmpA were revealed using SDS-PAGE (Appendix A). Compared with wild-type KJ, a band corresponding to 38 kDa was absent in the KJΔOmpA protein profiles, but was visible in KJL2::OmpAΔOmpA. The 38-kDa protein was verified as the OmpA protein using LC-MS/MS.

### 3.3. OmpA Deletion Attenuates Swimming Motility

The growth curves of wild-type KJ, KJΔOmpA, and KJL2::OmpAΔOmpA were determined during a period of 24 h, and were found to be undistinguishable. This indicated that *ompA* inactivation had no obvious effect on bacterial growth. 

Given that OmpA is the most abundant transcript revealed using transcriptome analysis, envelope integrity may be compromised in an OmpA protein-null mutant. An intact envelope is a prerequisite for flagellar assembly and swimming motility; thus, we were interested in understanding the impact of *ompA* deletion on swimming motility. The swimming motilities of KJ, KJΔOmpA, and KJL2::OmpAΔOmpA were assessed. Given that the complementary *ompA* gene in KJL2::OmpAΔOmpA was driven by the L2 promoter, a preliminary test was performed to determine the β-lactam concentration at which *ompA* can be expressed in the KJL2::OmpAΔOmpA. The swimming motility of KJL2::OmpAΔOmpA was tested with and without cefuroxime at different concentrations. Unexpectedly, KJL2::OmpAΔOmpA displayed comparable swimming motility in cefuroxime-free and cefuroxime-containing plates (Figure 2A), indicating that the *ompA* of KJL2::OmpAΔOmpA expressed by *P_L2_*, without β-lactam challenge, is sufficient to compensate for the functional defect caused by the *ompA* deletion. The mechanism underlying this phenotype will be explained later. Thus, the following functional experiments were carried out under β-lactam-free conditions. Compared to the 30 ± 3.2 mm swimming zone in wild-type KJ, a 13.6 ± 1.8 mm swimming zone was observed in KJΔOmpA (Figure 2A). The complementary strain KJL2::OmpAΔOmpA almost reverted the swimming motility to a wild-type level (Figure 2A). 

The flagellar structures of wild-type KJ, KJΔOmpA, and KJL2::OmpAΔOmpA were observed using transmission electron microscopy (TEM). Wild-type KJ displayed a tuft of 2–3 polar flagella (Figure 2B), which is consistent with previous reports [45]. In contrast, KJΔOmpA cells had short flagella that were sometimes located at a subpolar position. The complementary strain KJL2::OmpAΔOmpA restored the wild-type flagellar structure (Figure 2B). Next, flagellin protein levels were evaluated using a Western blotting assay. Considering the necessity of the FliC negative control for Western blotting, we aimed to construct a FliC-null mutant. In a genome-wide search, we noticed three flagellin-like genes in the *S. maltophilia* K279a genome, Smlt2306, Smlt2305, and Smlt2304. We annotated these three genes as *fliC1*, *fliC2*, and *fliC3*, respectively (Figure 3A). As the role of these three genes in flagellum formation is not yet understood, the flagellin-null mutant was constructed by simultaneously deleting the three genes to yield the mutant KJΔFliC1C2C3. The Western blotting results demonstrated that KJ, KJΔOmpA, and KJL2::OmpAΔOmpA had comparable cytoplasmic flagellin protein levels (Figure 2C).

### 3.4. Establishment of a Putative Flagellum Synthesis Model for S. maltophilia 

Information concerning flagellum synthesis and action in *S. maltophilia* is relatively scarce, and the hierarchical expression circuit has not been established. Although flagellum synthesis is well-studied in *E. coli* and *P. aeruginosa,* these systems are obviously different [27,28]. We surveyed the *S. maltophilia* K279a genome [40] for flagellum-associated genes and found that the flagellum system of *S. maltophilia* was more similar to that of *P. aeruginosa* [27]. To study the linkage between swimming compromise and *ompA* deletion, we thoroughly analyzed the *S. maltophilia* K279a genome to identify genes with known or predicted function of flagellum biogenesis or function, based on the known *P. aeruginosa* flagellum model. Genes known or predicted to be involved in chemotaxis and energy control were also included because of their genetic linkage to these flagellar genes. We identified 46 genes encoding the structural, assembly, or regulatory proteins of the flagellum in the sequenced K279a genome (Appendix A). Most of these genes have sufficient homology with their orthologs in *P. aeruginosa* (Appendix A). These genes are located in four clusters of chromosomes (Figure 3A). Most of the known flagellum-associated genes of *P. aeruginosa* are highly conserved in *S. maltophilia*, except for *fleRS, fleP,* and *fleL*.

A putative flagellum model of *S. maltophilia* was preliminarily proposed based on the *P. aeruginosa* model (Appendix A). FleQ and FliA are master regulators located in class I. FleQ works in concert with RpoN to activate the expression of class II genes. FlgM, a member of class II, interacts with FliA, resulting in FliA sequestration. In the *P. aeruginosa* model, class III genes are activated by class II-encoded FleRS in an RpoN-dependent manner [46]. However, no *fleRS* homologs have been identified in the *S. maltophilia* K279a genome. The regulatory components responsible for the expression of class III genes are relatively unclear. Upon completion of the hook basal body (HBB) by class II- and class III-encoded proteins, FlgM is secreted by the HBB and the liberated FliA activates the transcription of class IV genes.

To confirm the reliability of the preliminarily proposed flagellum model (Appendix A), we constructed isogenic deletion mutants of *fleQ* (Smlt2295), *rpoN* (Smlt2297), and *fliA* (Smlt2270), whose encoded proteins are predicted to function as regulators at the first hierarchical level of the *P. aeruginosa* flagellum model. None of the constructed mutants exhibited observable changes in their colony shapes and growth patterns. Wild-type KJ, KJΔFleQ, KJΔRpoN, and KJΔFliA were subjected to qRT-PCR analysis to confirm the accuracy of the hierarchical model. We randomly selected the following genes as representatives of the four classes of the proposed model (Appendix A) for qRT-PCR testing: *fleQ*, *rpoN*, and *fliA* for class I; *fliD* and *fliN* for class II; *flgG* and *flgK* for class III; and *fliC1, cheV*, and *motC* for class IV. The inactivation of *rpoN* reduced the expression of all the genes assayed, supporting RpoN as a class I regulator. *FleQ* deletion resulted in the downregulation of all the genes assayed except *rpoN*. Interestingly, *rpoN* was significantly upregulated in the *fleQ* mutant. The loss-of-function of *fliA* downregulated *fliC*, *cheV*, and *motC*, but upregulated *fliN*, *flgG*, and *flgK* (Figure 4A). The swimming motility, flagellum morphology, and flagellin protein levels of KJΔFleQ, KJΔRpoN, and KJΔFliA were also assessed. All of the mutants tested completely lost swimming motility (Figure 4B) and had no flagella (Figure 4C). This was further confirmed by the presence of few intracellular flagellin proteins (Figure 4D). Furthermore, the qRT-PCR results were validated using *xylE* reporter constructs in the *rpoN*, *fleQ*, and *fliA* mutant backgrounds. Using the plasmid pRpoN_xylE_ (a *P_rpoN_-xylE* transcriptional fusion), we measured *rpoN* promoter activity in the wild-type KJ, *fliA*, *rpoN*, and *fleQ* mutants. Compared to KJ(pRpoN_xylE_), KJΔFleQ(pRpoN_xylE_) had an increased C23O activity and KJΔFliA(pRpoN_xylE_) displayed a comparable one (Appendix A), consistent with the qRT-PCR results (Figure 4A). We also observed that no C23O activity was detected in KJΔRpoN(pRpoN_xylE_), indicating that RpoN has a positive autoregulation circuit. 

The flagellum hierarchy expression model of *S. maltophilia* was thus amended as follows (Figure 3B). RpoN is at the top of the regulatory hierarchy and drives the transcription of class II genes (labeled as a red oval in Figure 3B). Notably, the expression of *fliA* and *fleQ* was almost abolished in the *rpoN* mutant, indicating that *fleQ* and *fliA* are members of the RpoN regulon. Thus, *fleQ* and *fliA* were arranged at the class II level in the *S. maltophilia* model. This feature differs from the known model of *P. aeruginosa*. The proteins encoded by the class II genes included the flagellum component proteins (labeled as blue rectangles in Figure 3B) and regulatory proteins (labeled as blue ovals in Figure 3B). The key regulators for the expression of class III genes, such as FleSR of *P. aeruginosa*, are not immediately clear, but the qRT-PCR results supported the suggestion that RpoN and FleQ are involved in the expression of some class III genes. The proteins encoded by class III genes included flagellum component proteins, FlgBCDGHIKL (labeled as green rectangles in Figure 3B) and regulatory proteins (FliK and FlgD) involved in flagellum hook length and modification (labeled as green ovals in Figure 3B). After FlgM is secreted via HBB, free FliA switches the expression of class IV genes. The proteins encoded by class IV genes included flagellins, flagellum motor proteins, and chemotaxis proteins (labeled as purple rectangles in Figure 3B). In the following study, we applied the model proposed in Figure 3B as the rationale for the experimental design.

### 3.5. RpoN Downregulation Is Responsible for ΔompA-Mediated Swimming Compromise 

To identify the target genes responsible for swimming compromise in the *ompA* mutant, an RNA-Seq transcriptome assay of wild-type KJ and KJΔOmpA was conducted. A statistical significance was defined as an absolute fold change in RPKM equal to or greater than three. The transcriptome analysis revealed that 144 and 190 out of a total of 4695 genes were significantly increased and decreased, respectively, in KJΔOmpA compared to wild-type KJ (Appendix A). We noticed that the *L2* (Smlt3722) gene of KJΔOmpA showed a five-fold increase in the transcriptome assay (Appendix A), indicating that the *L2* gene is partially derepressed in KJΔOmpA. This observation provided a reasonable explanation for why the complementary strain KJL2::OmpAΔOmpA could revert to swimming motility without a β-lactam addition (Figure 2A). To validate the transcriptome results, qRT-PCR analysis was performed on the six selected genes. Relative to wild-type KJ, the expression of the six genes was corroborated by the transcriptome results in KJΔOmpA and mostly reverted to wild-type levels in KJL2::OmpAΔOmpA (Figure 5). 

As the first step in identifying the possible candidate genes responsible for the *ΔompA*-mediated swimming compromise, we thoroughly analyzed the transcriptome data (Appendix A), based on our proposed hierarchical regulation model for flagellum synthesis in *S. maltophilia* (Figure 3B). The surveyed genes included flagellum biosynthesis, flagellum regulation, flagellum motor, and chemotaxis (Table 1). Among the 46 genes surveyed, seven and four genes were significantly downregulated and upregulated in KJΔOmpA, respectively (Table 1, Figure 3A). The proteins encoded by the downregulated genes participated in global regulation (RpoN), flagellar assembly (FlhB, FliE, FliF, FliM, and FlgB), and motor switch control (FliL). The upregulated genes were *fliJ*, *fliK*, *flgK*, and *flgL*, whose encoded proteins are implicated in hook length control and hook assembly. According to the proposed hierarchy model in Figure 3B, RpoN is the key candidate responsible for *ΔompA*-mediated swimming compromise because it is located in the first class of the four-class model. 

To test this notion, KJΔOmpA was complemented with an intact *rpoN* gene. The *L2* gene was replaced with the *rpoN* gene using double cross-over recombination to yield KJL2::RpoN, and *ompA* was in-frame deleted from the chromosome of KJL2::RpoN, generating KJL2::RpoNΔOmpA (Figure 1). KJL2::RpoN displayed swimming motility comparable to that of wild-type KJ. However, compared to KJΔOmpA, KJL2::RpoNΔOmpA showed swimming motility restoration to the wild-type level (Figure 2A). The TEM analysis revealed that KJL2::RpoNΔOmpA displayed a wild-type flagellar structure (Figure 2B). The intracellular FliC protein level of KJL2::RpoNΔOmpA was also comparable to that of wild-type KJ (Figure 2C).

## 4. Discussion

For a pathogen, the first step in pathogenesis is to establish an initial attachment to the host cells and facilitate colonization. Colonization requires a functional flagellum; therefore, swimming motility is a crucial virulence factor for pathogens [47]. Although the flagellum and swimming motility are well characterized in several Gram-negative bacteria, such as *E. coli*, *V*. *cholerae*, and *P. aeruginosa*, little is known regarding these aspects in *S. maltophilia*. In this study, we first proposed the hierarchical organization of flagella synthesis genes in *S. maltophilia*. Many of the approximately 59 genes involved in flagella biosynthesis in *P. aeruginosa* are conserved in the *S. maltophilia* genome. However, some *P. aeruginosa* flagellar genes have not been identified in *S. maltophilia* (such as FleRS, FleP, and FleL), and some flagellar genes appear to be unique to *S. maltophilia* (such as Smlt2296). Furthermore, *P. aeruginosa* harbors a flagellin gene *(fliC)*, either type-a or type-b [48]; nevertheless, there are three flagellin genes (*fliC1*, *fliC2*, and *fliC3*) in the genome of *S. maltophilia* K279a.

In addition to the differences in gene organization, some regulatory mechanisms were noted to be different in the hierarchical models of *S. maltophilia* and *P. aeruginosa*, as follows: (i) Class I genes, *fleQ* and *fliA*, are constitutively expressed in *P. aeruginosa*. FleQ is considered the master regulator of the flagellar regulon [27] and works in concert with RpoN to trigger the expression of class II genes [49]. Furthermore, FliA is a critical regulator of class IV genes expression [49]. The transcription of *fleQ* and *fliA* is independent of RpoN in *P. aeruginosa* [50,51]. Nevertheless, in *S. maltophilia*, *rpoN* inactivation almost abolished the transcription of *fleQ, fliA,* and the genes of class II, class III, and class IV (Figure 3A), supporting the suggestion that *rpoN*, rather than *fleQ* and *fliA*, is at the top of the flagellar regulatory cascade in *S. maltophilia*. Thus, RpoN in *S. maltophilia* appears to have a more global regulatory spectrum than that in *P. aeruginosa*. Furthermore, RpoN has the feature of positive autoregulation (Appendix A). Therefore, the RpoN regulon of *S. maltophilia* encompasses the class I, II, III, and IV genes. (ii) The regulation of RpoN on *fleQ* did not appear to be a simple one-way regulatory circuit in *S. maltophilia*. The *rpoN* transcript was apparently upregulated in the *fleQ* mutant and was almost unexpressed in the *rpoN* mutant (Figure 3A and Appendix A). This phenomenon was not observed in *P. aeruginosa*. (iii) Similar to *P. aeruginosa* FliA, *S. maltophilia* FliA is the key regulator for the expression of class IV genes. Nevertheless, we observed that FliA of *S. maltophilia* can negatively regulate some class II and class III genes, thus extending our understanding of FliA. Interestingly, RpoN positively controls the expression of *fliA* and FliA negatively regulates the expression *of fliN*, *flgG*, and *flgK*; however, the expression of *fliN*, *flgG*, and *flgK* was almost abolished in KJΔRpoN (Figure 3A), suggesting the presence of unidentified regulator(s) participating in the complex regulatory circuit. 

OMPs can provide the structural integrity of bacteria, which is critical for flagellar assembly and swimming ability. The linkage between β-barrel OMPs and swimming motility has been reported in some bacteria, but the impact of OMP deletion on swimming motility is varied. The deletion of *ompX* increases swimming motility in extraintestinal pathogenic *E. coli* [52]. However, in *V. cholerae*, *flgO* and *flgP* deletion mutants show decreased swimming motility, and the stability of flagella is significantly affected [53]. In addition, Bari et al. have also reported that *ompU* and *ompT* deletion mutants of *V. cholerae* showed thinner flagella, significant defects in swimming motility, and increased shedding of flagella sheath proteins in the culture supernatant [54]. A similar observation was also reported for the *lamB* deletion mutant of *Aeromonas veronii*; its *lamB* deletion mutant almost lost swimming motility, and shedding flagella were found in the visual field of the TEM examination [55]. In this study, we demonstrated that the *ompA* deletion mutant of *S. maltophilia* has shorter flagella and defective swimming motility. Furthermore, we also revealed that *ΔompA*-mediated *rpoN* downregulation is a critical factor responsible for swimming compromise in KJΔOmpA. Precisely how *ompA* deletion downregulates *rpoN* expression remains unclear. The Gram-negative envelope protects the bacteria from the environment. The loss of abundant OmpA in the OM can cause envelope damage, which may trigger the envelope stress responses to maintain envelope homeostasis. The envelope stress responses extensively and diversely exist in different microorganisms, including σ^E^ response, Cpx response, Rcs response, Bae response, and Psp response [56]. We presume that the envelope stress response could be the connection between *ompA* deletion and *rpoN* downregulation. This issue needs to be further explored. 

The contribution of OmpA to bacterial conjugation has been reported in *E. coli* [57], and the underlying mechanism is that the TraN protein encoded by the F plasmid interacts with OmpA in recipient cells, thus increasing their mating efficiency [58]. In this study, we verified that *S. maltophilia* OmpA, similar to *E. coli* OmpA, is a critical protein in recipient cells for successful conjugation. Conjugation is instrumental to the horizontal transfer of antibiotic resistance genes in clinical environments and enhances the spread of antibiotic resistance among bacteria [59]. Thus, blocking conjugation can be regarded as a tool against antibiotic resistance [60].

*S. maltophilia* is ubiquitous in the environment and is an increasingly opportunistic pathogen. The attribute of multidrug resistance increases the difficulty of treating *S. maltophilia* infection [61]. Thus, it is important to develop new treatment strategies to combat *S. maltophilia* infections. Swimming is a critical virulence factor involved in bacterial pathogenesis. Conjugation is an efficient method for horizontal transfer of antibiotic resistance. In this study, we demonstrated that the OmpA of *S. maltophilia* significantly contributes to swimming and conjugation; thus, blocking the OmpA function may attenuate the bacterial pathogenicity and occurrence of antibiotic resistance. Therefore, OmpA is an ideal target for inhibitor design. For Gram-negative bacteria, the double membranes may be a natural barrier for the entrance of foreign molecules [2,3]. Given that OmpA is the OMP of the greatest abundance in *S. maltophilia*, the accessibility of an OmpA inhibitor to OmpA can be highly efficient. Therefore, the development of an OmpA inhibitor is a feasible strategy to combat *S. maltophilia* infection. 

## Figures and Tables

**Figure 1 microorganisms-09-01216-f001:**
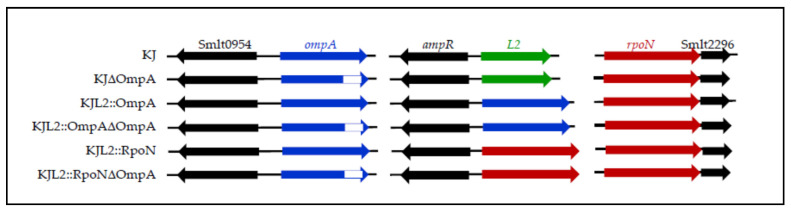
Genomic organization of KJ and its derived *ompA*-associated constructs. The orientation of the gene is indicated by the arrow. The white box indicates the deleted region of the *ompA* gene.

**Figure 2 microorganisms-09-01216-f002:**
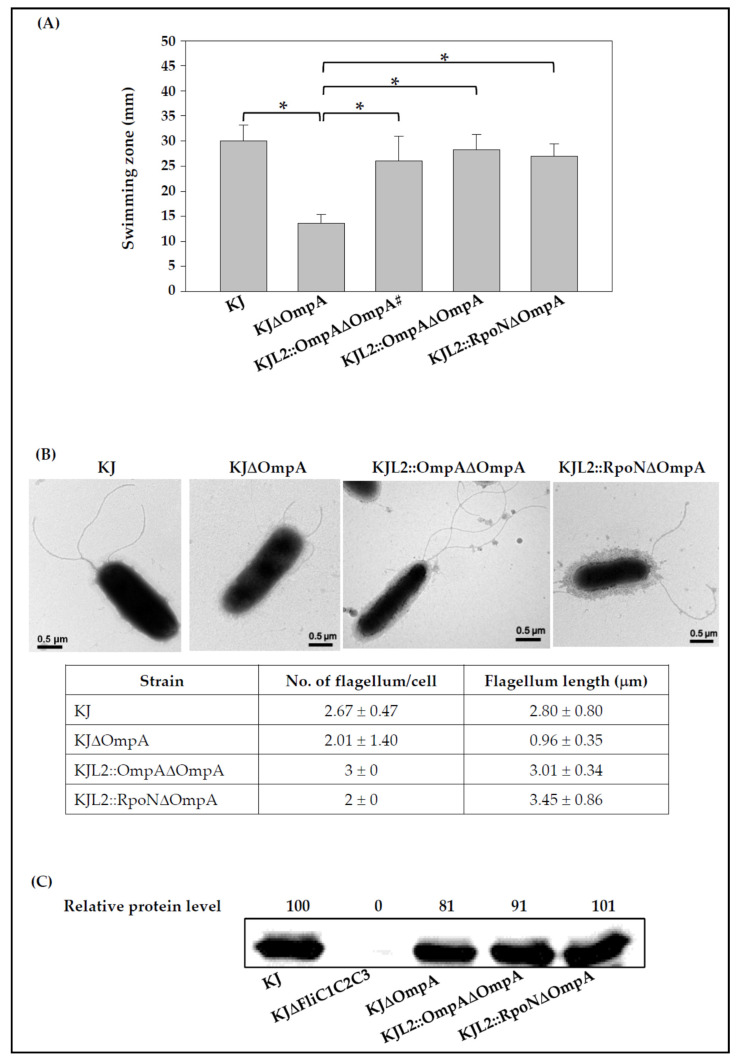
The swimming motility, flagellum morphology, and intracellular flagellin proteins levels of wild-type KJ, its derived *ompA* mutant, and complementary strains. Overnighted cultured bacterial cells were inoculated into fresh LB broth and then grown for 5 h before testing. (**A**) Swimming motility. Five μL of bacterial cell suspension were inoculated into swimming agar and then incubated at 37 °C for 48 h. The swimming zones were recorded. #, Cefuroxime of 50 μg/mL was added into the swimming agar. The data are the means of three independent experiments. The error bars indicate the standard deviations of three triplicate samples. *, *p* < 0.05, significance calculated using the Student’s *t*-test. (**B**) Flagella morphology. The flagella were negatively stained with 1% phosphotungstic acid (pH 7.4) and observed using TEM. The average flagellum numbers per cell and flagellum length were calculated from at least four cells. (**C**) Western blotting of flagellin protein levels in a whole-cell extract. The proteins of a whole-cell extract were separated using SDS-PAGE and immunoblotting with anti-FliC3 and anti-RpoA antibodies. The RpoA protein served as an intracellular control. The relative protein level of the FliC protein was normalized to the RpoA protein.

**Figure 3 microorganisms-09-01216-f003:**
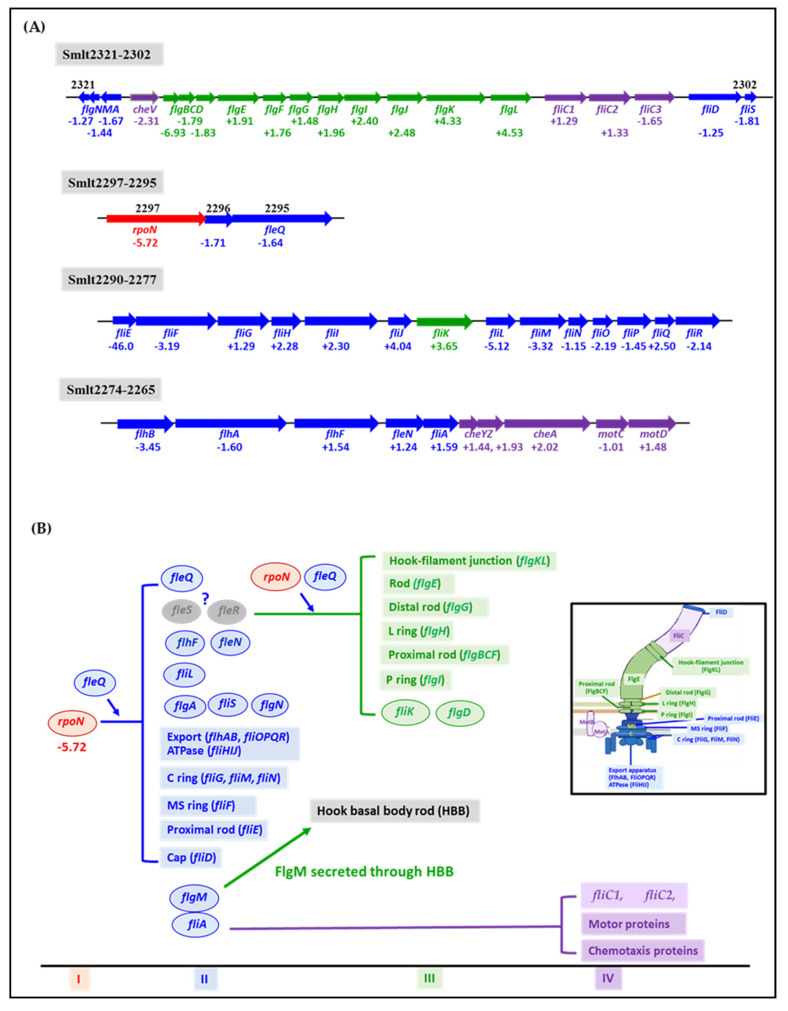
Flagellum synthesis model and RNA-seq transcriptome analysis of wild-type KJ and the *ompA* mutant, KJΔOmpA. (**A**) Genomic organization of the genes involved in flagellum synthesis and action. The flagellum-associated genes were located in four clusters. The genes classified in class I, II, III, and IV are marked in red, blue, green, and purple, respectively. The numbers below the genes indicate the gene expression changes analyzed using a transcriptome assay. Positive values represent an upregulation in KJΔOmpA, whereas negative values indicate a downregulation in KJΔOmpA. (**B**) A schematic model showing the transcriptional hierarchy of the genes involved in flagellum synthesis in *S. maltophilia.* The flagellum synthesis model of *S. maltophilia* is proposed by referencing the known *P. aeruginosa* flagellum model, as most of the flagellum synthesis-associated genes of *P. aeruginosa* are highly conserved in the *S. maltophilia* genome. The genes classified in class I, II, III, and IV are marked in red, blue, green, and purple, respectively. The genes symbolized by rectangles represent the encoded proteins composing the flagellum and the ovals represent regulatory proteins.

**Figure 4 microorganisms-09-01216-f004:**
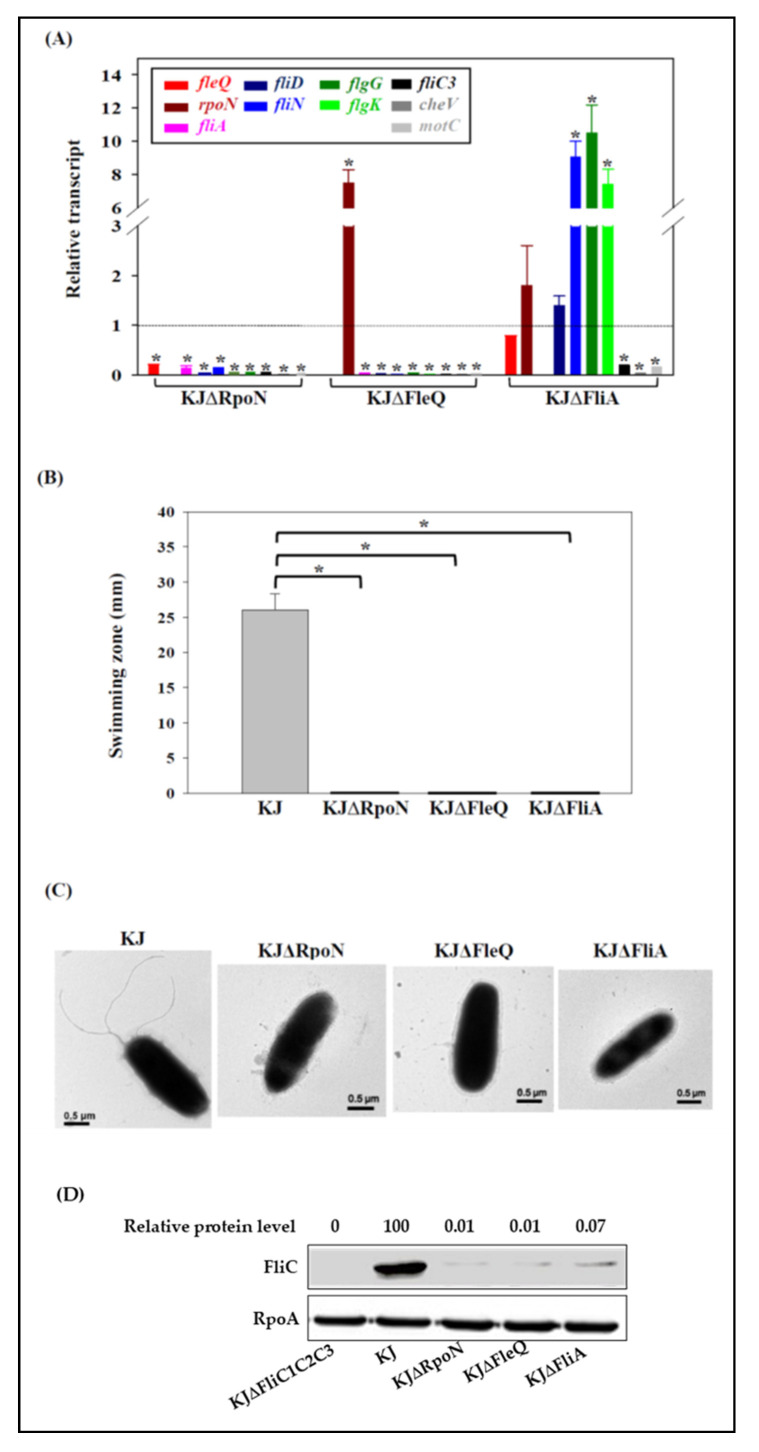
Impact of *rpoN*, *fleQ*, and *fliA* on flagellum-associated gene expression, swimming motility, flagellum morphology, and flagellin protein levels. (**A**) The impact of *rpoN*, *fleQ*, and *fliA* on flagellum-associated gene expression. Overnighted cultured bacterial cells (KJ, KJΔRpoN, KJΔFleQ, and KJΔFliA) were inoculated into fresh LB broth and then grown for 5 h before a qRT-PCR. The expression levels of the assayed genes were normalized to those of the 16 S ribosomal RNA gene. The relative transcript was calculated by dividing the mRNA levels of the wild-type by those of the mutant indicated. The mean and SD of the three independent experiments are shown. *, a relative transcript greater than 2 or less than 0.5 was considered to be significant. (**B**) The impact of *rpoN*, *fleQ*, and *fliA* on swimming motility. Overnighted cultured bacterial cells were inoculated into fresh LB broth and then grown for 5 h before testing. A five-microliter bacterial cell suspension was inoculated into swimming agar and then incubated at 37 °C for 48 h. The swimming zones were recorded. The data are the means of three independent experiments. The error bars indicate the standard deviations of three triplicate samples. *, *p* < 0.05, significance calculated using the Student’s *t*-test. (**C**) The impact of *rpoN*, *fleQ*, and *fliA* on flagella morphology. The flagella were negatively stained with 1% phosphotungstic acid (pH 7.4) and observed using TEM. (**D**) The impact of *rpoN*, *fleQ*, and *fliA* on flagellin protein levels. The proteins of a whole-cell extract were separated using SDS-PAGE and immunoblotting with anti-FliC3 and anti-RpoA antibodies.

**Figure 5 microorganisms-09-01216-f005:**
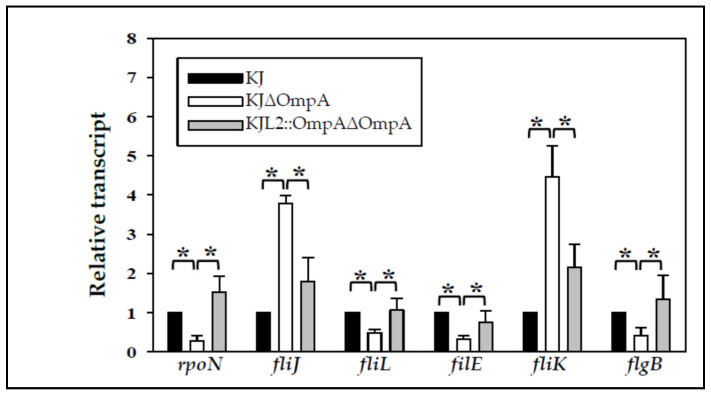
Impact of *ompA* on the expression of flagellum-associated genes. Overnighted cultured bacterial cells (KJ, KJΔOmpA, and KJL2:OmpAΔOmpA) were inoculated into fresh LB broth and then grown for 5 h before a qRT-PCR. The expression levels of the assayed genes were normalized to those of the 16 S ribosomal RNA gene. The relative transcript was calculated using the wild-type KJ level as one. The mean and SD of the three independent experiments are shown. *, *p* < 0.05, significance calculated using the Student’s *t*-test.

**Table 1 microorganisms-09-01216-t001:** Swimming-associated genes differently expressed in *S. maltophilia* KJ and KJΔOmpA.

Smlt	Gene	RPKM ^a^		Fold	Encoded Protein
		KJ	KJΔOmpA		
Class I					
2297	*rpoN*	104.48	18.26	**−5.72**	σ54 sigma factor
Class II					
2295	*fleQ*	226.20	137.15	−1.64	transcriptional activator
2270	*fliA*	98.17	156.68	+1.59	σ28 sigma factor
2271	*fleN*	159.21	197.77	+1.24	flagella number regulator
2272	*flhF*	14.26	22.09	+1.54	flagellar polar location
2273	*flhA*	8.14	5.07	−1.60	flagellar export protein
2274	*flhB*	8.63	2.49	**−3.45**	flagellar export protein
2277	*fliR*	14.57	6.78	−2.14	flagellar export protein
2278	*fliQ*	19.71	49.47	2.50	flagellar export protein
2279	*fliP*	15.18	10.43	−1.45	flagellar export protein
2280	*fliO*	38.14	17.37	−2.19	flagellar export protein
2281	*fliN*	52.03	45.24	−1.15	flagellar motor switch protein
2282	*fliM*	16.48	4.96	**−3.32**	flagellar motor switch protein
2283	*fliL*	19.49	3.80	**−5.12**	basal body-associated protein
2285	*fliJ*	12.00	48.53	**+4.04**	chaperone, export of hook proteins
2286	*fliI*	26.64	61.27	+2.30	flagellum-specific ATPase
2287	*fliH*	42.91	98.07	+2.28	flagella assembly protein
2288	*fliG*	56.09	72.82	+1.29	motor switch protein
2289	*fliF*	47.05	14.73	**−3.19**	basal body MS ring
2290	*fliE*	45.59	1.00	**−45.6**	basal body MS ring/rod adapter
2302	*fliS*	664.26	365.25	−1.81	chaperone for filament elongation
2303	*fliD*	297.89	237.50	−1.25	filament cap
2319	*flgA*	46.60	27.82	−1.67	basal body P-ring biosynthesis protein
2320	*flgM*	1908.60	1317.64	−1.44	anti-sigma factor FlgM
2321	*flgN*	775.79	607.01	−1.27	chaperone
Class III					
2284	*fliK*	19.85	72.54	**+3.65**	flagellar hook-length control
2307	*flgL*	54.28	246.26	**+4.53**	hook-filament junctional protein
2308	*flgK*	49.76	215.68	**+4.33**	hook-filament junctional protein
2309	*flgJ*	20.28	50.41	+2.48	flagellum specific muramidase
2310	*flgI*	15.17	36.478	+2.40	basal body P-ring
2311	*flgH*	36.78	72.24	+1.96	basal body L-ring
2312	*flgG*	60.97	90.60	+1.48	basal body rod
2313	*flgF*	5.34	9.46	+1.76	basal body rod
2314	*flgE*	13.12	25.11	+1.91	hook
2315	*flgD*	24.64	13.39	−1.83	hook capping protein
2316	*flgC*	15.23	8.49	−1.79	basal body rod
2317	*flgB*	48.91	7.05	**−6.93**	basal body rod
Class IV					
2265	*motD*	71.76	106.27	+1.48	flagellar motor protein
2266	*motC*	353.41	347.85	−1.01	flagellar motor protein
2267	*cheA*	92.22	186.55	+2.02	chemotaxis sensor kinase regulator
2268	*cheZ*	241.78	467.49	+1.93	chemotaxis protein
2269	*cheY*	268.99	388.99	+1.44	chemotaxis response regulator
2304	*fliC1*	395.14	510.14	+1.29	flagellin
2305	*fliC2*	23.32	31.10	+1.33	flagellin
2306	*fliC3*	180.47	109.26	−1.65	flagellin
2318	*cheV*	164.29	70.89	−2.31	chemotaxis response regulator

^a^ RPKM, Reads Per Kilobase of transcript per Million reads mapped. Bold: Statistical significance was defined as an absolute fold change in RPKM equal to or greater than 3.

## Data Availability

Not applicable.

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
