# Peer review of "Interplay between OmpA and RpoN Regulates Flagellar Synthesis in Stenotrophomonas maltophilia"

_microorganisms, 2021, doi:10.3390/microorganisms9061216_

Round 1
Reviewer 1 Report
The authors investigated the effect of the loss-of-ompA and showed that the outer membrane protein is involved in synthesis of flagella through association with the flagellum-related gene rpoN. The data obtained by transcriptome and qRT-PCR are used for assessing how the in-frame-deletion affect the flagellar synthesis, which will be useful for deeper understanding of the gene regulation in the future. My concerns are listed below.
(1) Fig 1B is the most important in this paper, but the authors show only one image for each strain. It is mandatory to present quantitative data, which is given by measuring the number and length of flagella.
(2) If Fig 1B is right, the effect of -ompA on swimming is indirect one. Therefore, I suggest changing the title, e.g., Interplay between OmpA and RpoN regulates flagellar synthesis in Stenotrophomonaas maltophilia. If the authors suspect the effect on swimming, they should observe swimming under microscope and measure swimming speed at least. Table 1 shows that chemotaxis-related genes are upregulated in the ompA deficient mutant, suggesting that chemotaxis (i.e., reversal frequency of flagellar rotation and swimming direction) might be affected by lacking ompA. If such phenotypes will be confirmed, the present title will be appropriate.
Author Response
Comments and Suggestions for Authors
The authors investigated the effect of the loss-of-ompA and showed that the outer membrane protein is involved in synthesis of flagella through association with the flagellum-related gene rpoN. The data obtained by transcriptome and qRT-PCR are used for assessing how the in-frame-deletion affect the flagellar synthesis, which will be useful for deeper understanding of the gene regulation in the future. My concerns are listed below.
- Fig 1B is the most important in this paper, but the authors show only one image for each strain. It is mandatory to present quantitative data, which is given by measuring the number and length of flagella.
Reply: The flagellum numbers and flagellum length have been included. Please see the Fig. 2B in the revised manuscript.
(2) If Fig 1B is right, the effect of -ompA on swimming is indirect one. Therefore, I suggest changing the title, e.g., Interplay between OmpA and RpoN regulates flagellar synthesis in Stenotrophomonaas maltophilia. If the authors suspect the effect on swimming, they should observe swimming under microscope and measure swimming speed at least. Table 1 shows that chemotaxis-related genes are upregulated in the ompA deficient mutant, suggesting that chemotaxis (i.e., reversal frequency of flagellar rotation and swimming direction) might be affected by lacking ompA. If such phenotypes will be confirmed, the present title will be appropriate.
Reply: Thank you very much for your suggestions. I totally agreed your opinions. The title has been modified accordingly. Please see Lines 2-3 in the revised manuscript. Thanks again.

Reviewer 2 Report
The manuscript „Interplay between OmpA, RpoN, and swimming motility in Stenotrophomonas maltophilia” is a significant contribution as the emerging resistance rates of this pathogen are concerning, and studies associated with the virulence factors of this bacterium are always welcome. The study has been done with care taken to modern molecular biological methods and careful interpretation of the results.
Here are the concerns need addressing:
L21: based on trascriptome analyses.
L26: The flagellum is a critical organelle for swimming.: this sentence should either be complemented or removed
L39-44: please discuss that this double membrane may be impenetrable to antibiotics or other noxius agents
Please include the follwoing reference:
https://www.ncbi.nlm.nih.gov/pmc/articles/PMC7355843/
https://pubmed.ncbi.nlm.nih.gov/30832456/
L42: Gram-negative
I suggest the introduction of the abbreviation OM for outer membrane
L52: Gram-negative
L53: use instead the order name (Enterobacterales)
L80-83: please discuss that motility may be varied based on the QS signals in the environment
L85: Gram-negative
- maltophilia is considered a low-virulence pathogen; however, the treatment of S. maltophilia infection is a major challenge due to its intrinsic resistance to several antibiotics that are commonly used in clinic.
Please include the following reference:
https://doi.org/10.1556/1886.2020.00006
L93: Please include the following reference:
https://aac.asm.org/content/64/9/e00559-20
L97: Gram-negative
L99-100: please corrent Li et al.
Section 2.1. I suggest to present some information on the origin, number, ID methods of these isolates!!!
Results:
would it be possible to summarize the results in sections 3.2. and 3.3. in the form of figures?
L550: Gram-negative
L609: S. maltophilia is ubiquitous in the environment and is an increasingly opportunistic pathogen.
Please include the following reference, and elaborate on the clinical syndromes where S. maltophilia may be a relevant clinical pathogen:
https://doi.org/10.1177/2333392819870774
L608-L616: please elaborate more on this topic, as this paragraph feels unfinished.
Author Response
Comments and Suggestions for Authors
The manuscript „Interplay between OmpA, RpoN, and swimming motility in Stenotrophomonas maltophilia” is a significant contribution as the emerging resistance rates of this pathogen are concerning, and studies associated with the virulence factors of this bacterium are always welcome. The study has been done with care taken to modern molecular biological methods and careful interpretation of the results.
Here are the concerns need addressing:
L21: based on trascriptome analyses.
Response: The word, analyses, has been corrected accordingly. Please see Line 21 in the revised manuscript.
L26: The flagellum is a critical organelle for swimming.: this sentence should either be complemented or removed
Response: The sentence has been removed.
L39-44: please discuss that this double membrane may be impenetrable to antibiotics or other noxius agents
Please include the follwoing reference:
https://www.ncbi.nlm.nih.gov/pmc/articles/PMC7355843/
https://pubmed.ncbi.nlm.nih.gov/30832456/
Response: Thank you very much for your suggestion. The concept of double membranes and references have been added. Furthermore, the viewpoints of double membranes barrier were also included in the Discussion section. Please see Lines 40-41, 627-634, and reference 2 & 3 in the revised manuscript.
L42: Gram-negative
Response: The word, Gram, has been corrected accordingly. Please see Line 43 in the revised manuscript.
I suggest the introduction of the abbreviation OM for outer membrane
Response: The abbreviation OM for outer membrane has been used all through the manuscript.
L52: Gram-negative
Response: The word, Gram, has been corrected accordingly. Please see Line 53 in the revised manuscript.
L53: use instead the order name (Enterobacterales)
Response: The word, Enterobacterales, has been corrected accordingly. Please see Line 54 in the revised manuscript.
L80-83: please discuss that motility may be varied based on the QS signals in the environment
Response: The concept of QS signals has been added. Please see Line 84 in the revised manuscript.
L85: Gram-negative
Response: The word, Gram, has been corrected accordingly. Please see Line 87 in the revised manuscript.
L86: S. maltophilia is considered a low-virulence pathogen; however, the treatment of S. maltophilia infection is a major challenge due to its intrinsic resistance to several antibiotics that are commonly used in clinic.
Please include the following reference:
https://doi.org/10.1556/1886.2020.00006
Response: The reference has been added. Please see Line 91 and reference 30 in the revised manuscript.
L93: Please include the following reference:
https://aac.asm.org/content/64/9/e00559-20
Response: The reference has been added. Please see Line 94 and reference 32 in the revised manuscript.
L97: Gram-negative
Response: The word, Gram, has been corrected accordingly. Please see Line 98 in the revised manuscript.
L99-100: please corrent Li et al.
Response: It has been corrected accordingly. Please see Line 102 in the revised manuscript.
Section 2.1. I suggest to present some information on the origin, number, ID methods of these isolates!!!
Response: The information about the isolate has been added. Please see Lines 114-115 in the revised manuscript.
Results:
would it be possible to summarize the results in sections 3.2. and 3.3. in the form of figures?
Response: The results of section 3.2 were summarized as Figure 1 in the revised manuscript. The results of section 3.3 were integrated into the Figure 2A in the revised manuscript.
L550: Gram-negative
Response: The word, Gram, has been corrected accordingly. Please see Line 564 in the revised manuscript.
L609: S. maltophilia is ubiquitous in the environment and is an increasingly opportunistic pathogen.
Please include the following reference, and elaborate on the clinical syndromes where S. maltophilia may be a relevant clinical pathogen:
https://doi.org/10.1177/2333392819870774
Response: The reference has been added. Please see Line 624 and reference 60 in the revised manuscript.
L608-L616: please elaborate more on this topic, as this paragraph feels unfinished.
Response: The more detailed discussion has been included. Please see Lines 627-634 in the revised manuscript.

Reviewer 3 Report
The ms describes the validation of a target in S.malt hop hilos.
the results are of interest to the readers and scientific community, however more solid conclusions and discussion are needed. Which next step to do? Etc etc
Author Response
The ms describes the validation of a target in S. maltophilia.
the results are of interest to the readers and scientific community, however more solid conclusions and discussion are needed. Which next step to do? Etc etc
Response: The more discussions have been added. Please see Lines 611-618 and 632-639 in the revised manuscript.
Round 2
Reviewer 1 Report
I acknowledge that the authors addressed my comments. I request to describe the sample number measured in Fig 2B (the number and length of flagella).
Author Response
I acknowledge that the authors addressed my comments. I request to describe the sample number measured in Fig 2B (the number and length of flagella).
Reply: The description of sample numbers has been included. Please see the Fig. 2B (Line 349) in the revised manuscript.
Reviewer 3 Report
The ms is now acceptable for publication